# Printed Soft Sensor with Passivation Layers for the Detection of Object Slippage by a Robotic Gripper

**DOI:** 10.3390/mi11100927

**Published:** 2020-10-08

**Authors:** Reo Miura, Tomohito Sekine, Yi-Fei Wang, Jinseo Hong, Yushi Watanabe, Keita Ito, Yoshinori Shouji, Yasunori Takeda, Daisuke Kumaki, Fabrice Domingues Dos Santos, Atsushi Miyabo, Shizuo Tokito

**Affiliations:** 1Research Center for Organic Electronics (ROEL), Graduate School of Science and Engineering, Yamagata University, 3-4-16, Jonan, Yonezawa, Yamagata 992-8510, Japan; tnh99114@st.yamagata-u.ac.jp (R.M.); wang@yz.yamagata-u.ac.jp (Y.-F.W.); jinseo.hong@yz.yamagata-u.ac.jp (J.H.); tnn26268@st.yamagata-u.ac.jp (Y.W.); tae33543@st.yamagata-u.ac.jp (K.I.); txc00320@st.yamagata-u.ac.jp (Y.S.); y.takeda@yz.yamagata-u.ac.jp (Y.T.); d_kumaki@yz.yamagata-u.ac.jp (D.K.); 2Piezotech S. A. S., Arkema-CRRA, Rue Henri Moissan, 63493 Pierre-Benite Cedex, France; fabrice.domingues-dos-santos@arkema.com; 3Arkema K. K., 2-2-2 Uchisaiwaicho, Chiyoda-ku, Tokyo 100-0011, Japan; atsushi.miyabo@arkema.com

**Keywords:** wearable sensing, soft sensor, robotics, shear force, dynamic friction coefficient

## Abstract

Tactile sensing, particularly the detection of object slippage, is required for skillful object handling by robotic grippers. The real-time measurement and identification of the dynamic shear forces that result from slippage events are crucial for slip detection and effective object interaction. In this study, a ferroelectric polymer-based printed soft sensor for object slippage detection was developed and fabricated by screen printing. The proposed sensor demonstrated a sensitivity of 8.2 μC·cm^−2^ and was responsive to shear forces applied in both the parallel and perpendicular directions. An amplifier circuit, based on a printed organic thin-film transistor, was applied and achieved a high sensitivity of 0.1 cm^2^/V·s. Therefore, this study experimentally demonstrates the effectiveness of the proposed printable high-sensitivity tactile sensor, which could serve as part of a wearable robotic e-skin. The sensor could facilitate the production of a system to detect and prevent the slippage of objects from robotic grippers.

## 1. Introduction

Currently, industrial robots provide automated systems that can grasp and manipulate objects and serve in fields such as electronic device manufacturing and food industries [1,2]. Robots with soft sensors for tactile sensing have recently attracted considerable research attention. Functionalized flexible sensors that enable sensing, such as touch, temperature, and vision [3,4,5,6,7,8,9,10] in robot systems, have been applied to enhance automated systems [11,12,13,14,15]. Moreover, because robots with such sensors can be applied to artificial intelligence and big data, their usefulness has been expanded [16,17]. In particular, the detection of object slippage from robotic grippers by using the fabricated soft sensor is an important issue for robotic system control. 

Piezoelectric polymer-based tactile slippage detection sensors can measure the shear force generated by a slippage event. Recent studies have demonstrated the use of fabricated ferroelectric-material-based sensors for robotic e-skin [18,19,20]. Soft sensors can be mounted on rigid- and soft-robot for making a sensing system such a biomimetic tactile sensor. To date, however, no studies have reported on the use of printed soft sensors to sense the shear forces experienced by robotic grippers. The arrangement and types of materials for the fabrication of these high-performance devices have remained a challenging task [21]. Therefore, it is necessary to improve the sensitivity of ferroelectric materials for shear force detection. Further, a printing process for device fabrication would be superior to current sensor manufacturing systems because it would enable the use of highly efficient materials and the production of low-cost devices. Recently, S. Pyo et al. reported a Carbon nanotube (CNT) -polymer composite based tactile sensor. In this paper, they developed a flexible three-axis tactile sensor by screen printing a system with a carbon nanotube-polymer materials composite. The CNT-based sensor had an arrayed-pattern and can measure an applied stress by changing resistance [22]. In addition, this sensor can be fabricated by screen printing, which has been widely used to fabricate various micropatterns for its cost effectiveness. J. Hwang et al., fabricated a pressure sensor using piezoresistive composites by multi-walled carbon nanotubes and polydimethylsiloxane as a polymer matrix. This can detect an extremely small pressure range which is required for finger-sensing [23]. On the other hand, the piezoelectric polymer-based sensor does not require an array-pattern. Moreover, it can detect a shear force that is changing in minute time.

In this study, a ferroelectric polymer-based printed soft sensor was fabricated to sense and measure the dynamic shear forces generated during a slippage event. The proposed sensor achieved a sensitivity of 8.2 μC·cm^−2^, and a robotic gripper that included the sensor demonstrated the ability to detect shear force. Moreover, an amplifier circuit using an organic thin-film transistor (OTFT), which was also fabricated using a printing process, was applied and achieved a high sensitivity of 0.1 cm^2^/V·s. The results demonstrate the feasibility of producing a robotic gripper that includes a high-sensitivity electronic skin system for detecting slippage events.

## 2. Materials and Methods

### 2.1. Device Fabrication

A schematic overview of the fabricated soft shear sensor is shown in Figure 1a,b. The sensor was fabricated on a 50 μm thick (Q65HA, DuPont) poly(ethylene naphthalate) (PEN) film substrate and affixed to a glass carrier. A cross-linkable poly(4-vinyl-phenol) (PVP) (436224, Sigma-Aldrich) solution consisting of a mixture of PVP and melamine resin (418560, Sigma-Aldrich) was spin-coated onto the PEN film as the planarization layer using 1-methoxy-2-propyl acetate (01948-00, Kanto Chemicals) as the solvent. An 800 nm thick lower electrode made from poly(3,4-ethylenedioxythiophene) polystyrene sulfonate (PEDOT:PSS) (Clevios SV4 STAB, Heraeus) was screen printed onto the planarization layer and annealed at 150  °C for 30 min [24]. Further, a 8000 nm thick layer of poly(vinylidene fluoride-co-trifluoroethylene) (P(VDF-TrFE)) (FC-25, Piezotech, VDF:TrFE molar ratio of 75:25) dissolved in trimethylphosphate (TMP) at a concentration of 12 wt.% was formed by blade printing and annealed at 145  °C for 1 h. The upper sensor electrode consisted of screen printed PEDOT:PSS and was annealed at 145  °C for 30  min. Finally, several passivation layers consisting of PEN, polyimide (PI), and polyvinyl chloride (PVC) were placed onto the device. The total weight of our sensor was just under 1.0 g. The chemical structures of the P(VDF-TrFE) and the components of the passivation layers are shown in Figure 1c.

### 2.2. Sensor Characterization

All the sensor characteristics were measured under atmospheric conditions using an oscilloscope (MDO3000, Tektronix) and a waveform generator (AFG3101C, Tektronix). The polarization–electric field (P–E) loop of the sensor was estimated using the Sawyer–Tower method [25]. The sensor piezoelectricity was measured using an electric slider testing machine (EASM4NYD010AZAC, Oriental Motor). X-ray diffraction (XRD) (SmartLab, Rigaku) measurements of the P(VDF-TrFE) layer were performed to assess crystallinity and analyze the crystal structures. The surface and interface morphologies of the P(VDF-TrFE) layer were observed using an atomic force microscope (AFM) (5500, Agilent). Fourier transform infrared (FT-IR) measurements were performed at a spectral resolution of 2 cm using an FT-IR spectrometer (Nicolet iS5, Thermo Scientific). The dynamic friction coefficient of the devices was evaluated using a dynamic friction measuring instrument (µV 1000, Trinity Lab). 

### 2.3. Amplifier Circuit

An amplifier circuit was developed using an OTFT to achieve high sensitivity. The device was fabricated using bottom-gate and bottom-contact techniques. The OTFT consisted of a flexible substrate (PEN), bottom electrode (deposited Al, 30 nm), insulator (polyparaxylene, 200 nm), source–drain (S/D) electrodes (Ag nanoparticles, 500 nm), semiconductors, and passivation layers (polyparaxylene, 200 nm). The length and width of the OTFT channels were approximately 30 and 800 μm, respectively. In this case, the thickness of PVDF-based ferroelectric layer used was 2000 nm.

## 3. Results and Discussion

Figure 2 shows the electrical and chemical characteristics of the sensor. Figure 2a shows the P–E loop providing the ferroelectric polarization of the sensor. The measured polarization, *P_r_* (µC·cm^−2^), and coercive electric field, *E_c_* (MV·m^−1^), were 8.2 µC·cm^−2^ and 50 MV·m^−1^, respectively, which are considered appropriate for a P(VDF-TrFE)-based printed ferroelectric layer [26]. Figure 2b shows a surface AFM image of the printed P(VDF-TrFE) layer. The Root Mean Square (RMS) value calculated from its surface morphology was 15.0 nm. Figure 2c shows the 2θ peak signal of the printed P(VDF-TrFE) layer, which indicated a (110/200) crystal face, highlighted by the purple dashed box. This result clearly demonstrated that the sensor had a *β* phase in this layer [27]. Figure 2d provides the FT-IR spectra of the printed ferroelectric layers in powdery, the signal peaks at 1405 and 845 cm^−1^ represent the *β* phase transformation of the P(VDF-TrFE) [28]. Figure 2e illustrates the dependence of the sensor polarization on the annealing temperature; the average values for the three samples are shown. Annealing at 145 °C provided the highest polarization value of 8.2 µC·cm^−2^ because the crystallization temperature of the P(VDF-TrFE) was in the range of 130–150 °C [29].

The dynamic coercive friction values of the passivation sensor layers were also measured, as shown in Figure 3. The measurement setup of the dynamic friction measuring instrument consisted of a load-cell, indenter (glass ball), stage, and the soft sensor samples, which were attached to the stage (Figure 3a). The instrument employed in the dynamic friction measurements obtained the friction values by the indenter scanning across the surface of the sample in the parallel direction (Figure 3b); the perpendicular force of the indenter on the sample was 1 N. Figure 3c shows the measured friction coefficients for the PEN, PI, and PVC layers as a function of the scanning distance; the coefficients were determined to be 0.12, 0.18, and 0.24, respectively. The reported values were obtained by averaging the results for several distances between 10 and 50 mm. The dependence of the obtained friction coefficient values upon the indenter scanning speed was tested for speeds of 1–5 mm·s^−1^, and the results are illustrated in Figure 3d,e. The raw data measured for the PEN-covered sensor are shown in Figure 3d. Figure 3e provides the friction coefficient values as a function of the scanning speed for the different passivation layer materials. From these results, the friction coefficient did not depend on scanning speed.

Figure 4 shows the experimental setup for determining the slip-sensing abilities of the sensor. Figure 4a provides a photo image of the measurement setup with the artificial finger, which was moved over the object surface by the electric slider; therefore, the force *F* acted in the parallel direction. Figure 4b illustrates the friction model developed from the equation of motion. The model was calculated by the formulae representing the frictional force [29].

The sensor generated voltage in response to the dynamic friction caused by the object slippage. Because the sensor was composed of a piezo-material, the voltage generation phenomena are described by [30]
(1)$$V=ad_{33}\frac{d\sigma }{dt}\;T\;(4),$$
where *a* is a coefficient, *d*_33_ is the piezoelectric constant, σ is the applied strain, *t* is time, and *T* is the thickness of the P(VDF-TrFE) layer. Figure 4c shows the sensor response upon the application of a parallel force. In this test, the passivation layers were the PEN, PI, and PVC films. When the artificial finger contacted and moved over the object surface, the sensor produced a piezoelectrical signal that resulted in a voltage spike (the green and orange arrows in Figure 4c). In contrast, a periodic signal was generated when a parallel (shear) force was applied (the purple arrow in Figure 4c). Figure 4d shows the measurement setup with the robotic gripper. The friction model developed based on the equation of motion is shown in Figure 4e. This model was calculated based on the formula of friction force as follows [29]:
(2)$$F_{1}-F_{2}=0;$$
(3)$$f_{1}+f_{2}=mg.$$

The force *mg* acts in the perpendicular direction. Figure 4f shows the sensor response when the object fell from the robotic gripper. For this test, the passivation layers of the sensor were also the PEN, PI, and PVC films. As in the previous test, a voltage spike was generated when the gripper contacted and scanned the object surface. Figure 4g magnifies the graph in Figure 4f for the PEN film passivation layer to illustrate the response when the object slipped. Periodic stick-slip (S-S) signals were generated during the slip phase due to the jerking motion between the two slipping objects (i.e., the soft sensor and object) [30,31,32]. Moreover, the spectra repeated periodically if the object fell at a constant velocity. Appendix A shows the mechanical fatigue of our sensor as a generated voltage during long-term cycling. Moreover, in Appendix A, we described the occurrence of shear force. The S-S spectra of the voltage features exhibited time-dependent variations, as shown in Figure 4e. A glass bottle with a low frictional coefficient produced S-S signals with a frequency of ~100 Hz. In this work, a quick-time response is required in the soft shear sensor because the S-S phenomena occurred in minute time. Our sensor could measure the S-S slippage events in real-time monitoring. We demonstrated the detection of objects slippage using our sensor which has high sensitivity and quick-time response. In addition, through layer-by-layer production on the substrate via printing technology, the sensor was flexible and lightweight. Thus, we could detect slip events by using our sensor mounted on the robot hands. In particular, the characteristic of quick-time response of the sensor was useful for finding the S-S phenomena between the object and robot hands.

An amplifier circuit was developed using an organic thin-film transistor (OTFT) to achieve high sensitivity. Figure 5a provides a cross-sectional schematic of the fabricated OTFT device. In general, slippage signals tend to be weak. Therefore, we composed the amplifier circuit with OTFT devices which can produce flexible electronics. Moreover, we demonstrated a connection system with the sensor and OTFT in Appendix A. Figure 5b shows the chemical structure of the molecule selected to serve as the organic semiconductor. Figure 5c provides a diagram of the amplifier circuit and sensor. In this circuit, when pressure was applied to the sensor, voltage was generated to the gate electrode, and the difference in current between the S/D electrodes was measured. The transfer characteristics of the OTFT—as an amplification circuit component—are displayed in Figure 5d. The applied gate and drain voltages were −30 V. The field-effect mobility of the OTFTs was 0.1 cm^2^/V·s in the saturation region. Figure 5e,f show the results of applying perpendicular pressure to the sensor at 1 kPa and 10 kPa, respectively; the current changes were then analyzed (shown by the green arrows in Figure 5e,f). We succeeded in detecting the applied pressure by using our soft sensor and OTFT device. From the results, the use of OTFTs as amplifier components for the soft sensors was demonstrated to be feasible.

## 4. Conclusions

In this study, a soft sensor comprising a ferroelectric polymer and several passivation layers was developed and fabricated to sense object slippage. P(VDF-TrFE) served as the ferroelectric layer, and PEN, PI, and PVC films were the passivation layers. This ferroelectric-based sensor demonstrated good ferroelectric characteristics, providing ferroelectricity of 8.2 μC·cm^−2^ after annealing at 140 °C. The device was proven to be practical for detecting object slippage events, based on its reactions to parallel and perpendicular forces. Moreover, the OTFT devices were demonstrated to be practical amplifiers, by demonstrating stable electric performance and a field-effect mobility of 0.1 cm^2^/V·s. From the results, the use of OTFTs as amplifier components for the soft sensors was demonstrated to be feasible. Moreover, we showed the potential application of OTFT as an amplifier circuit for sensor devices. The work in this study could serve as a foundation for the development of effective slippage detection sensors for application in robotic gripper devices. This study tested only one proposed device and fabrication technique. Future work could provide comparative evaluations to ensure the highest performance and greatest cost-effectiveness.

## Figures and Tables

**Figure 1 micromachines-11-00927-f001:**
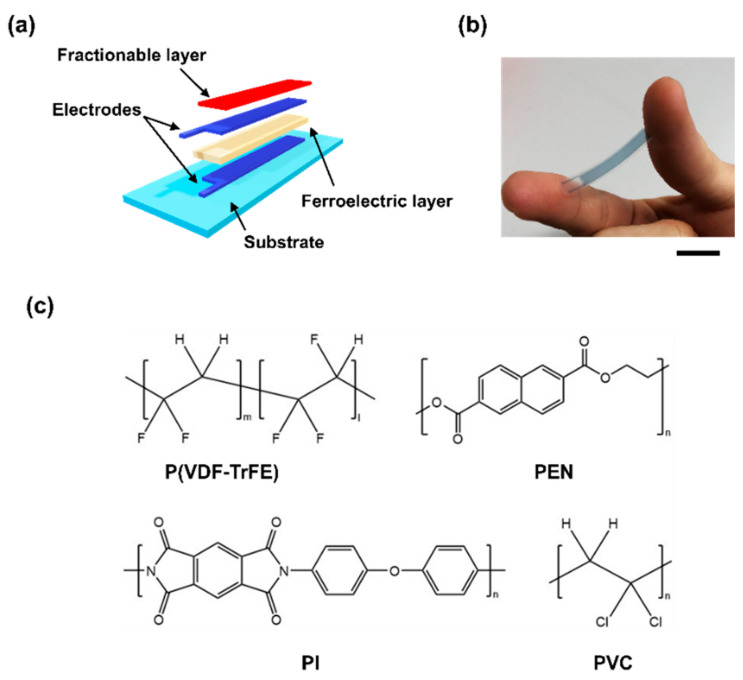
Soft shear force sensor fabrication: (**a**) Illustration of the fabricated sensor; (**b**) Photo of the sensor, scale bar = 5 cm; (**c**) Chemical structures of the ferroelectric and passivation layer materials. poly(vinylidene fluoride-co-trifluoroethylene): (P(VDF-TrFE)); poly(ethylene naphthalate): (PEN); polyimide: (PI); polyvinyl chloride: (PVC).

**Figure 2 micromachines-11-00927-f002:**
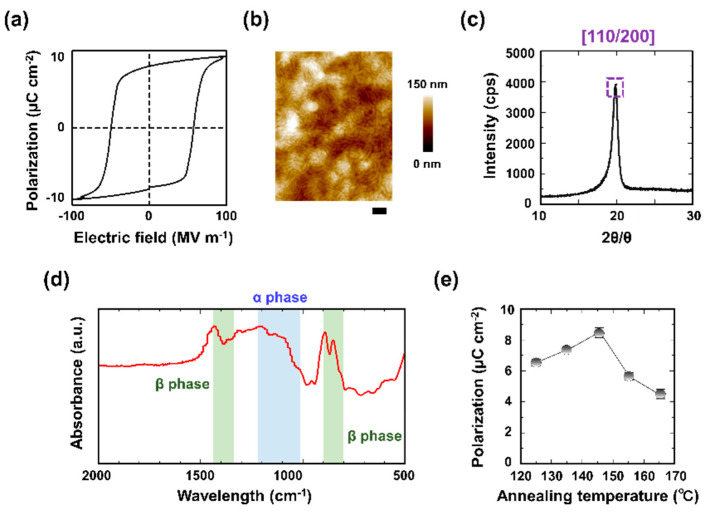
Electrical and chemical characteristics of the soft shear force sensor: (**a**) Measured P-E curve (1 Hz sensor frequency); (**b**) Surface atomic force microscope (AFM) images of the P(VDF-TrFE) layers annealed at 145 °C (scale bar = 100 nm); (**c**) Printed ferroelectric layer XRD spectra after annealing; (**d**) Ferroelectric layers FT-IR spectra; (**e**) Ferroelectric layer polarization as a function of the annealing temperature.

**Figure 3 micromachines-11-00927-f003:**
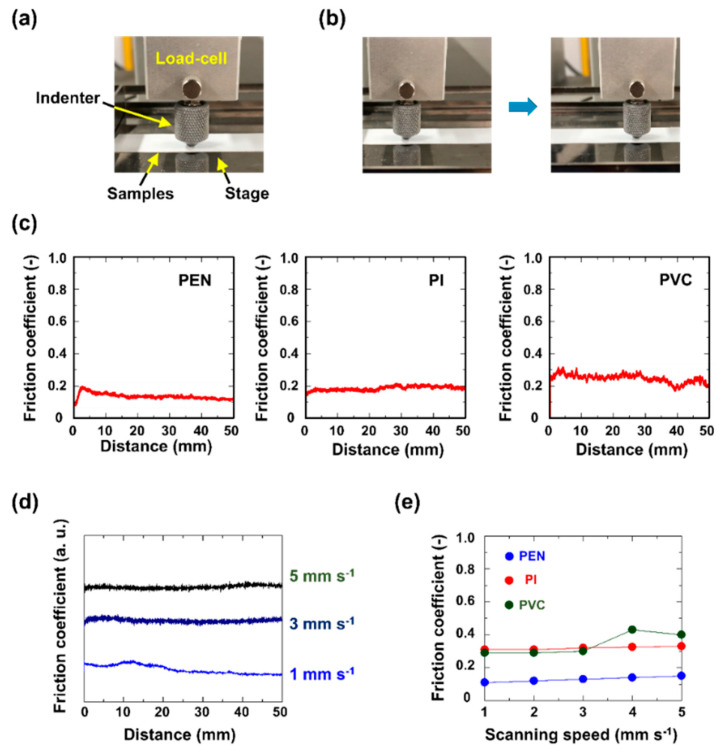
Dynamic friction coefficient measurements: The sensor layers comprised PEN, PI, and PVC films: (**a**) Measurement setup; (**b**) Indenter scanning to perform the measurements; (**c**) Measured friction coefficients of the fractional layers; (**d**) Friction coefficients as a function of the scanning speeds; (**e**) Friction coefficients of the fractional layers as a function of the scanning speeds.

**Figure 4 micromachines-11-00927-f004:**
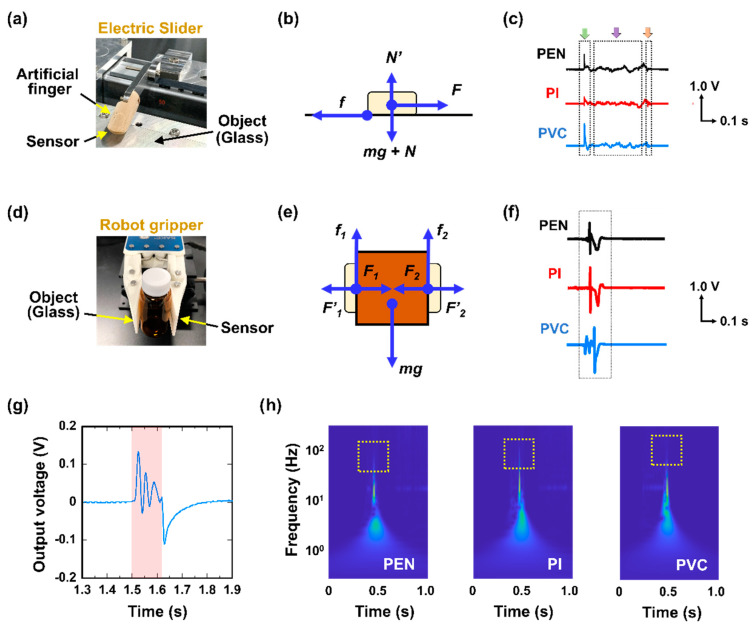
Object slippage detection by the soft sensor: (**a**) Measurement setup showing the artificial finger and the electric slider; (**b**) Friction model based on the equation of motion; (**c**) Signals detected by the sensors in each passivation layer (PEN, PI, and PVC); (**d**) Measurement setup with the robotic gripper; (**e**) Friction model developed from the equation of motion; (**f**) Signals detected by the robotic gripper sensors for each passivation layer (PEN, PI, and PVC); (**g**) Voltage response of the PEN film during object slippage (magnification of (f)). The stick-slip (S-S) spectra are shown in red; (**h**) S-S spectra of the piezoelectric voltage signals for different fractional layers over a range of frequencies.

**Figure 5 micromachines-11-00927-f005:**
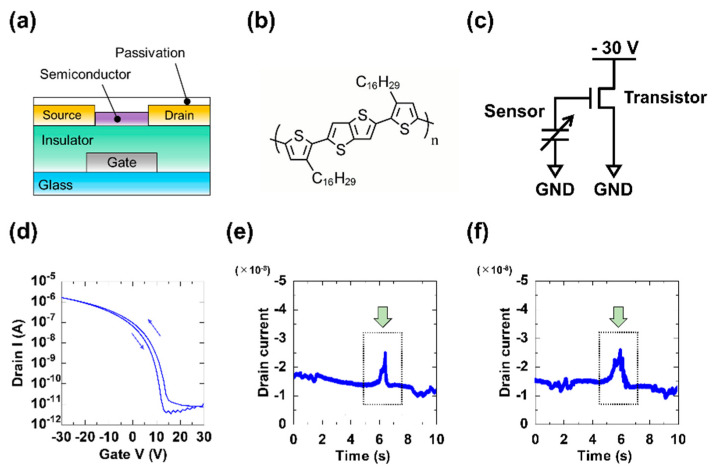
An organic thin-film transistor (OTFT) device: (**a**) Fabricated OTFT device; (**b**) Chemical structure of the molecule intended as the organic semiconductor; (**c**) Amplifier circuit diagram; (**d**) Transfer characteristics of the OTFT; Piezoelectric responses of the OTFT device under (**e**) 1 kPa and (**f**) 10 kPa of applied pressure.

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
