# Peer review of "Printed Soft Sensor with Passivation Layers for the Detection of Object Slippage by a Robotic Gripper"

_micromachines, 2020, doi:10.3390/mi11100927_

Round 1
Reviewer 1 Report
This paper presents a tactile sensor design, used for normal and shear stress measurement. The novelty, as claimed by the author, resides in the fact that the sensor (and its amplifier) are both printed. However, there are several concerns with this paper, some of which are noted below:
- Minor: in the abstract, the authors mention that the sensor is responsive to shear forces in parallel and perpendicular directions. What exactly does that mean and particularly, what are supposed to be "perpendicular shear forces"? Please make sure to use the right terms, e.g. Normal stress and shear stress.
- Major: Just on the basis of the introduction quality, this paper could be rejected. The authors didn't provide a real introduction, the contribution and novelty is not clear. Also, the author claims "To date, however, no studies have reported on the use of printed soft sensors to sense the shear forces [...]" -> This is false, see for example "Development of a flexible three-axis tactile sensor based on screen-printed carbon nanotube-polymer composite" by Pyo et al.
- Major: There is no real state-of-the-art / literature review in this paper. The authors simply enumerated a bunch of broad (i.e. non-specific) references in their short introduction. Sometimes, the references are cited in agglomerations of up to 8. Nothing specific is said about any comparable work, no comparison is provided to highlight what are the important contributions from this work. Printed soft tactile sensors are not new.
- The "Materials and Methods" section is interesting, especially the part about the device fabrication.
- Minor: All figures should be a bit bigger, especially the graphs.
- Moderate: In section 2.2 - "Sensor Characterization" the authors should at least briefly explain why all these characteristics are important to measure. For example, why is it important to measure the piezoelectricity of the sensor? As an experience tactile sensor user for more than 10 years, it is not clear to me why the authors cared to measure all of these characteristics. Also, if so many characteristics were measured, it would have been great to measure the capacitance of the sensor (capacitance of the electrodes under no normal stress, and a curve showing the evolution of the capacitance for different normal stress values).
- Neutral: it would have been great to make a distinction between incipient slip and gross slip. It would have been interesting to see if the proposed sensor could be used to detect incipient slip before gross slip happens.
- Moderate: it's very difficult to see the experimental setup from figure 4a.
- Major: Reference [27], which is cited for equations (1) and (2) is not useful. Please don't add a broad list of papers for general statements which add absolutely no value to the paper.
- Major: Equations (1), (2) and (3) are way too elementary to put in a scientific paper. They add nothing to the paper.
- Major: I would agree that this sensor is different from other tactile sensors found in the literature. However, it's not clear at all how this different tactile sensor would be useful to any practical context. The authors should improve the way they explain why this contribution is important, what are the merit of this sensor over existing tactile sensing technologies (i.e. to make a better litterature review, among other things).
- Minor: There are unit inconsistencies found in the paper: cm^2*V*s is not adequate and is present at different locations in the paper.
- Major: the slippage detection experiment is too minimalist to be considered a conclusive experiment on the performance of this sensor to detect slippage.
- Additional comment: in the tactile sensing field, it's consider relatively easy to build a soft sensor which is able to react when slippage occurs. How is this sensor better, or which are the characteristics that make it important to share with other researchers? This question should be answer more adequately in a future version of the paper.
Author Response
#_Reviewer 1
This paper presents a tactile sensor design, used for normal and shear stress measurement. The novelty, as claimed by the author, resides in the fact that the sensor (and its amplifier) are both printed. However, there are several concerns with this paper, some of which are noted below:
Minor: in the abstract, the authors mention that the sensor is responsive to shear forces in parallel and perpendicular directions. What exactly does that mean and particularly, what are supposed to be "perpendicular shear forces"? Please make sure to use the right terms, e.g. Normal stress and shear stress.
------We appreciate your instructive comments. As you mentioned, we should make sure shear stress in this paper. As the followed reply, we defined the direction of several stress as normal and shear.
Major: Just on the basis of the introduction quality, this paper could be rejected. The authors didn't provide a real introduction, the contribution and novelty is not clear. Also, the author claims "To date, however, no studies have reported on the use of printed soft sensors to sense the shear forces [...]" -> This is false, see for example "Development of a flexible three-axis tactile sensor based on screen-printed carbon nanotube-polymer composite" by Pyo et al.
------ The authors agreed the reviewer’s comments. In the current manuscript, it did not provide a clear introduction to the contribution in a shear force sensor. By Pyo et al. studied “Flexible three-axis tactile sensor by carbon nanotube-polymer material with screen printing system”. By comparing with this is paper, our sensor does not require an array-pattern. Moreover, it can detect a shear force changing in minute time. We revised the mechanism of the micropores generation in the main manuscript in page 1, line 43 as the following;
(Revised sentences)
Recently, S. Pyo et al. reported a CNT-polymer composite based tactile sensor. In this paper, they developed a flexible three-axis tactile sensor by screen printing system with carbon nanotube-polymer materials composite. The CNT-based sensor had an arrayed- pattern and can measure an applied stress by changing resistance [22]. In addition, this sensor can be fabricated by screen printing, which has been widely used to fabricate various micropatterns for its costeffectiveness. J. Hwang et al., fabricated a pressure sensor using piezoresistive composites by multi-walled carbon nanotubes and polydimethylsiloxane as a polymer matrix. It can detect an extremely small pressure range which required for finger-sensing [23]. On the other hand, the piezoelectric polymer-based sensor does not require an array-pattern. Moreover, it can detect a shear force changing in minute time.
Major: There is no real state-of-the-art / literature review in this paper. The authors simply enumerated a bunch of broad (i.e. non-specific) references in their short introduction. Sometimes, the references are cited in agglomerations of up to 8. Nothing specific is said about any comparable work, no comparison is provided to highlight what are the important contributions from this work. Printed soft tactile sensors are not new. The "Materials and Methods" section is interesting, especially the part about the device fabrication.
------ The authors would agree to this comment. In current manuscript, a little description of printed shear sensors is in main text. Therefore, we added new sentences which explain novel points of our sensor by comparing with previous reports as the following in page 1, line 43;
(Revised sentences)
Recently, S. Pyo et al. reported a CNT-polymer composite based tactile sensor. In this paper, they developed a flexible three-axis tactile sensor by screen printing system with carbon nanotube-polymer materials composite. The CNT-based sensor had an arrayed- pattern and can measure an applied stress by changing resistance [22]. In addition, this sensor can be fabricated by screen printing, which has been widely used to fabricate various micropatterns for its costeffectiveness. J. Hwang et al., fabricated a pressure sensor using piezoresistive composites by multi-walled carbon nanotubes and polydimethylsiloxane as a polymer matrix. It can detect an extremely small pressure range which required for finger-sensing [23]. On the other hand, the piezoelectric polymer-based sensor does not require an array-pattern. Moreover, it can detect a shear force changing in minute time.
Minor: All figures should be a bit bigger, especially the graphs.
------Thank you very much for your comment. We revised the size of all figures in the main manuscript.
Moderate: In section 2.2 - "Sensor Characterization" the authors should at least briefly explain why all these characteristics are important to measure. For example, why is it important to measure the piezoelectricity of the sensor? As an experience tactile sensor user for more than 10 years, it is not clear to me why the authors cared to measure all of these characteristics. Also, if so many characteristics were measured, it would have been great to measure the capacitance of the sensor (capacitance of the electrodes under no normal stress, and a curve showing the evolution of the capacitance for different normal stress values).
------In this study, our sensor is based on piezoelectric. In the sensor based on that principle, the piezoelectricity such P-E curve can show sensing characteristics. Maybe as the reviewer know, wider P-E curve show higher sensing ability for applied strain. Moreover, in piezoelectric devices, there is reliability between the polarization value and the capacitance value.
Neutral: it would have been great to make a distinction between incipient slip and gross slip. It would have been interesting to see if the proposed sensor could be used to detect incipient slip before gross slip happens.
------As the reviewer said, it has been interesting to see if the proposed sensor could be used to detect incipient slip before gross slip happens. For the next step, we should continue to analyze the detected slippage data against to classify slippage state clearer.
Moderate: it's very difficult to see the experimental setup from figure 4a.
------We appreciate your kindly indication. We changed the experimental setup in Fig. 4a.
Major: Reference [27], which is cited for equations (1) and (2) is not useful. Please don't add a broad list of papers for general statements which add absolutely no value to the paper.
------According to this comment, we deleted Ref. [27] from the main text.
Major: Equations (1), (2) and (3) are way too elementary to put in a scientific paper. They add nothing to the paper.
------ According to this comment, we deleted Equations (1), (2) and (3) from the main text.
Major: I would agree that this sensor is different from other tactile sensors found in the literature. However, it's not clear at all how this different tactile sensor would be useful to any practical context. The authors should improve the way they explain why this contribution is important, what are the merit of this sensor over existing tactile sensing technologies (i.e. to make a better litterature review, among other things).
------We appreciate your instructive comment. As the reviewer said, we revised and added sentences to deeply understand in the experimental section in page 7, line 277 as the following;
(Added sentences)
We demonstrated the detection of objects slippage using our sensor which has high sensitivity and quick-time response. In addition, making layer-by-layer on the substrate with printing technology, the sensor was flexible and lightweight. Thus, we could detect slip events by using our sensor mounted on the robot hands. Especially, the characteristic of quick-time response of the sensor was useful for finding of S-S phenomena between the object and robot hands.
Minor: There are unit inconsistencies found in the paper: cm^2*V*s is not adequate and is present at different locations in the paper.
------Thank you very much for pointing out. These are our typo. We revised the unit as “cm2/Vs”.
Major: the slippage detection experiment is too minimalist to be considered a conclusive experiment on the performance of this sensor to detect slippage.
------As the reviewer said, we should try more experiments against deeply understanding. In Fig. S1, we added a mechanical fatigue of our sensor as a generated voltage during long-term cycling. From this result, the sensor has the long-term stability.
Additional comment: in the tactile sensing field, it's consider relatively easy to build a soft sensor which is able to react when slippage occurs. How is this sensor better, or which are the characteristics that make it important to share with other researchers? This question should be answer more adequately in a future version of the paper.
------ We appreciate your instructive comment. One of the caricaturists of our sensor is high sensitivity and quick-time response by printing technology. Especially, effect of the fractionable layer for detected signals is also new discovery in this study. In near future, we’re going to research relation of the above in printed ferroelectric sensors.

Reviewer 2 Report
The authors present a novel slip sensor, using a piezoelectric approach and fabricated through a screen printing method.
The work in the paper is sound, high quality and very relevant to the journal readership; slip sensing is a key area of need in robotics and other fields.
I have some reservations with the current manuscript which omits some details necessary to provide context to the work, to justify the approach and to highlight the contributions made by this work (with respect to the literature). These reservations could be readily addressed through a revision of the manuscript, the technical work itself is of high quality and uses an impressive array of techniques to develop the work.
A version of the manuscipt is provided with full comments and areas to address marked up for authors' attention. In summary, the key areas which I believe require attention are:
- definition of the term 'fractionable' and emphasis of why this is a desirable property. Consider if this is the best term to use for a more general readership
- INTRO: define why soft sensors are useful for slip sensing
- INTRO: Provide context to justify the selection of PE based slip sensing vs other modalities
- INTRO: provide an overview of the sensor concept and justification
- Improve the clarity in several figures (see PDF) - and ensure full descriptions are provided in the captions.
- METHODS: you do not evaluate the mechanical properties of the sensor - you need to comment on how this qualifies as a 'soft' sensor, providing some information to justify this description. Do you mean flexible, deformable, conformable?
- RESULTS: Improve the rigour in describing the application of slip sensing - how measured signals relate to the force/events occuring.
- RESULTS: The amplifier circuit should be described in methods.
- RESULTS: Explain why this amplifier was necessary (vs alternatives) - what benefits does it bring?
- RESULTS: can you show a depiction of the full system; sensor + amplifier
- CONCs: More consideration of wider literature is required, in particular to emphasise the contributions of this work.

Author Response
#_Reviewer 2
The authors present a novel slip sensor, using a piezoelectric approach and fabricated through a screen printing method. The work in the paper is sound, high quality and very relevant to the journal readership; slip sensing is a key area of need in robotics and other fields. I have some reservations with the current manuscript which omits some details necessary to provide context to the work, to justify the approach and to highlight the contributions made by this work (with respect to the literature). These reservations could be readily addressed through a revision of the manuscript, the technical work itself is of high quality and uses an impressive array of techniques to develop the work.A version of the manuscipt is provided with full comments and areas to address marked up for authors' attention. In summary, the key areas which I believe require attention are:
definition of the term 'fractionable' and emphasis of why this is a desirable property. Consider if this is the best term to use for a more general readership
------Thank you very much for positive and instructive comments. As the reviewer said, the word of “fractionable” is few confusions for paper readers. The authors think maybe a word of “passivasion” is better than current word. In main text, we changed as the above. Moreover, the checklist of attached PDF was also useful for our revision. Many thanks for the reviewer’s efforts.
INTRO: define why soft sensors are useful for slip sensing
------Thank you very much for your question. Our soft sensor can be mounted on rigid- and soft-robot for making sensing system. These advantages are added in the main text in page 1, line 37 as the following;
(Revised sentence)
Soft sensors can be mounted on rigid- and soft-robot for making sensing system such a biomimetic tactile sensor.
INTRO: Provide context to justify the selection of PE based slip sensing vs other modalities
------ We agree to the reviewer. In this work, quick-time response is required in the soft shear sensor because S-S phenomena occurred in minute time. Our sensor could measure the S-S slippage events in real-time monitoring. The above description was added in main text in page 7, line 253 as the following;
In this work, quick-time response is required in the soft shear sensor because S-S phenomena occurred in minute time. Our sensor could measure the S-S slippage events in real-time monitoring.
INTRO: provide an overview of the sensor concept and justification
------We agree to the reviewer. In page 1, line 43 in main text, we overviewed the sensor concept and justification compering with previous reports as the following;
(Added sentences)
Recently, S. Pyo et al. reported a CNT-polymer composite based tactile sensor. In this paper, they developed a flexible three-axis tactile sensor by screen printing system with carbon nanotube-polymer materials composite. The CNT-based sensor had an arrayed-pattern and can measure an applied stress by changing resistance [22]. In addition, this sensor can be fabricated by screen printing, which has been widely used to fabricate various micropatterns for its costeffectiveness. J. Hwang et al., fabricated a pressure sensor using piezoresistive composites by multi-walled carbon nanotubes and polydimethylsiloxane as a polymer matrix. It can detect an extremely small pressure range which required for finger-sensing [23]. On the other hand, the piezoelectric polymer-based sensor does not require an array-pattern. Moreover, it can detect a shear force changing in minute time.
Improve the clarity in several figures (see PDF) - and ensure full descriptions are provided in the captions.
------Thank you very much for your kindly effort. We checked that and revised point-by-point.
METHODS: you do not evaluate the mechanical properties of the sensor - you need to comment on how this qualifies as a 'soft' sensor, providing some information to justify this description. Do you mean flexible, deformable, conformable?
------The authors agree to your comment. We should show mechanical properties of our sensor against applied strains. In this revision state, as one of the tryouts, a mechanical fatigue of our sensor as a generated voltage during long-term cycling in Fig. S1. According to this work, our sensor has enough mechanical property as soft sensors for soft robotics.
RESULTS: Improve the rigor in describing the application of slip sensing - how measured signals relate to the force/events occurring.
------We appreciate your kindly comment. The description of shear force occurring is attached in Fig. S2.
RESULTS: The amplifier circuit should be described in methods.
------We added description for the amplifier circuit in the section of “Materials and Methods” as the following;
(Added sentences)
An amplifier circuit was developed using an OTFT to achieve high sensitivity. The device was fabricated using bottom-gate and bottom-contact techniques. The OTFT consisted of a flexible substrate (PEN), bottom electrode (deposited Al, 30 nm), insulator (polyparaxylene, 200 nm), source-drain (S/D) electrodes (Ag nanoparticles, 500 nm), semiconductors, and passivation layers (polyparaxylene, 200 nm). The length and width of the OTFT channels were approximately 30 and 800 μm, respectively. In this case, the thickness of PVDF-based ferroelectric layer of 2000 nm was used.
RESULTS: Explain why this amplifier was necessary (vs alternatives) - what benefits does it bring?
------ In general, slippage signals have a tendency to be weak. Therefore, we consisted the amplifier circuit with OTFT devices which can realize flexible electronics. Moreover, we showed a connection system with the sensor and OTFT in Fig. S3. In page 7. line 278, the above was added as the following;
(Added sentences)
In general, slippage signals tend to be weak. Therefore, we consisted the amplifier circuit with OTFT devices which can realize flexible electronics. Moreover, we showed a connection system with the sensor and OTFT in Fig. S3.
RESULTS: can you show a depiction of the full system; sensor + amplifier
------We added detail information of the full system; sensor + amplifier in Fig. S3.
CONCs: More consideration of wider literature is required, in particular to emphasise the contributions of this work.
------We appreciate your kindly comment. According to the reviewer’s indication, we added new description in the sanction of conclusion as the following;
(Added sentences)
From the results, the use of OTFTs as amplifier components for the soft sensors was demonstrated to be feasible. Moreover, we showed potential application of OTFT as an amplifier circuit for sensor devices.
Other
------The reviewer attached other comments in PDF file. We appreciate your deeply effort. Also, we checked that and revised point-by-point. We believe that the current style appropriates to publication for this journal. In particular, image quality, size and visibility of some image are revised according to the comments.

Round 2
Reviewer 1 Report
The authors have sufficiently addressed the comments made by the reviewers.